# Salt Tolerance of *Hydrangea* Plants Varied among Species and Cultivar within a Species

**Genhua Niu** [1,*] **, Youping Sun** [2] **, Triston Hooks** [1] **, James Altland** [3] **, Haijie Dou** [1,4] **and Christina Perez** [1,5]

1   Texas A&M AgriLife Research, Texas A&M University, Dallas, TX 75252, USA;
    triston.hooks@ag.tamu.edu (T.H.); haijiedou@outlook.com (H.D.); cperez86@nmsu.edu (C.P.)
2   Department of Plants, Soils, and Climate, Utah State University, Logan, UT 84322, USA;
    youping.sun@usu.edu
3   USDA-ARS, Wooster, OH 44691, USA; james.altland@usda.gov
4   Department of Science & Technology Development, Beijing Industrial Technology Research Institute,
    Tongzhou District, Beijing 101111, China
5   Department of Plant and Environmental Sciences, New Mexico State University, Las Cruces, NM 88003, USA
*   Correspondence: gniu@ag.tamu.edu

**Abstract:** A greenhouse study was conducted to assess the relative salt tolerance of 11 cultivars of hydrangea: *Hydrangea macrophylla* 'Ayesha', 'Emotion', 'Mathilda Gutges', 'Merritt's Supreme' and 'Passion'; *H. paniculata* 'Interhydia' and 'Bulk'; *H. quercifolia* 'Snowflake'; *H. serrata* 'Preciosa'; and *H. serrata* × *macrophylla* 'Sabrina' and 'Selina'. Plants were treated with a nutrient solution at an electrical conductivity (EC) of 1.0 dS·m$^{-1}$, and nutrient solution-based saline solutions at an EC of 5.0 dS·m$^{-1}$ (EC 5) or 10 dS·m$^{-1}$ (EC 10). The study was repeated in time (Experiments 1 and 2). In both experiments, by the fourth week after treatment, 'Bulk' plants in EC 10 exhibited severe salt damage with most of them dead. 'Interhydia' was also sensitive, showing severe salt damage in EC 10 with a high mortality rate by the end of the experiment. The leaf area and total shoot dry weight (DW) of all cultivars in EC 5 and EC 10 treatments were significantly reduced compared to the control. Leaf sodium (Na$^+$) and chloride (Cl$^-$) concentrations were negatively correlated with visual quality, leaf area and shoot DW. The salt-sensitive cultivars 'Bulk', 'Interhydia' and 'Snowflake' had inherently low leaf Na$^+$ and Cl$^-$ concentrations in both control and salt-treated plants compared to other cultivars. Salt tolerance varied among species and cultivars within *H. macrophylla*. Among the 11 cultivars, *H. macrophylla* 'Ayesha' and two hybrids, 'Sabrina' and 'Selina', were relatively salt-tolerant. *H. macrophylla* 'Merritt's Supreme' and 'Mathilda' were moderately tolerant. *H. paniculata* 'Bulk' was the most sensitive, followed by *H. paniculata* 'Interhydia', and then by *H. serrata* 'Preciosa' and *H. macrophylla* 'Passion', as evidenced by high mortality and severe salt damage symptoms. *H. quercifolia* 'Snowflake' and *H. macrophylla* 'Emotion' were moderately salt-sensitive.

**Keywords:** nursery crops; ornamentals; saline water irrigation; salt tolerance mechanism

## 1. Introduction

The availability of high-quality water for agriculture and landscape irrigation has decreased due to several reasons, including drought, increasing population, climate change and depletion of aquifers. The intense competition for high-quality water among agriculture, industry and other users is promoting the use of alternative water sources for irrigating crops such as nursery crops and landscapes [1–4], specialty medicinal plants [5], and food crops such as vegetables [6]. Alternative waters are nontraditional waters, including treated municipal reclaimed (recycled) water, runoff from

greenhouse and nursery operations, agricultural drainage water, and naturally-occurring saline groundwater. The primary challenges of using water from these alternative sources are the high salt levels and undesirable specific ions, particularly sodium ($Na^+$) and chloride ($Cl^-$), which are ubiquitous in both soil and water and may be harmful to plant growth at elevated levels. Soil salinization tends to be common in arid and semi-arid regions where the leaching of salts through the soil profile is poor due to low rainfall. Furthermore, the use of alternative water sources with high salinity exacerbates soil salinization.

Hydrangea, native to Asia and the Americas, is one of the most popular and widely grown and marketed plants in the nursery and floriculture industry, owing to its beautiful flowers that come in a variety of shapes, colors and sizes. There are numerous species of deciduous and evergreen hydrangea shrubs, small trees and climbers, which include approximately 1000 cultivars and hybrids [7]. Many new selections are introduced annually worldwide due to its increasing popularity.

A lot of research has been focused on hydrangea breeding [8,9] and production [10,11]; however, limited information is available on the salt tolerance of hydrangea, particularly among different species and cultivars. Wu and Dodge [12] reported that *H. macrophylla* (no cultivar was indicated) was tolerant to salt spray between 200 mg $L^{-1}$ sodium and 400 mg $L^{-1}$ chloride, and moderately tolerant to soil salinity at an electrical conductivity (EC) of 2 to 4 dS·$m^{-1}$. Miralles et al. [2] investigated the responses of *H. macrophylla* 'Leuchtfeuer' (pink flower) irrigated with saline reclaimed water at an EC of 5.65 dS·$m^{-1}$ or with fresh water at an EC of 0.87 dS·$m^{-1}$. Their findings indicated that the elevated salts in saline reclaimed water severely decreased the size of all the aerial organs, including leaf, inflorescence and flower tissues, reduced the dry weight of the aerial parts by 70%, delayed flowering, and produced a darker pink floral color compared to the plants irrigated with fresh water. Liu et al. [13] evaluated the morphological and physiological responses of 10 woody ornamental taxa, including two *H. macrophylla* cultivars 'Smnhmsigma' and 'Smhmtau', to saline water irrigation at an EC of 5 or 10 dS·$m^{-1}$ for 8 weeks. The two cultivars of *H. macrophylla* were among the most salt-tolerant taxa with minor foliar damage; however, they had the highest leaf sodium and chloride concentrations, indicating that *H. macrophylla* plants adapted to elevated salinity by tolerating high ion concentrations in the tissue.

Hydrangeas are popular landscape plants in coastal areas. In order to aid in the selection for seaside planting, Conolly et al. [14] conducted a study to quantify the responses of five hydrangea species to foliar salt spray with full (ion concentration approximate to seawater at 35 parts per thousand) or half-strength sodium chloride solution, in seven once-weekly applications. Based on percentage necrotic leaf area, they concluded that *H. macrophylla* and *H. serrata* were more tolerant to foliar salt spray than *H. paniculata*, *H. anomala* and *H. arborescens*, and could be planted where maritime salt spray occurs.

Considering the above studies, there is evidence that some cultivars of hydrangea may be salt-tolerant, and that the tolerance may be species-specific. It is known that the salt tolerance of crops varies among species, and even cultivars within a species, and has been observed in many ornamentals such as aster [15] and roses [16]. Therefore, the purpose of this study was to evaluate the relative salt tolerance of 11 popular commercial hydrangea cultivars based on their responses to elevated salinity levels in their growth, visual quality and leaf mineral contents, particularly the uptake of sodium and chloride in leaves.

## 2. Materials and Methods

### 2.1. Plant Materials

Softwood cuttings of 11 hydrangea cultivars (Table 1) were received on 31 May 2017 from the Oregon Hydrangea Company (Brookings, OR, USA). Cuttings were treated with 1000 mg $L^{-1}$ indole-3-butyric acid (IBA) (1.0% IBA, 0.5% 1-napthaleneacetic acid, 98.5% inert ingredients (Dip'N Grow® liquid rooting concentrate; Dip'N Grow®, Clackamas, OR, USA)) following a quick-dip technique. They were planted in 9 cm pots containing a mixture of perlite and Metro-Mix® 360 RX

(SunGro, Bellevue, WA, USA) at a volumetric ratio of 1:1. Cuttings were misted on a bench for 15 s every 30 min using an automated controller (Trident T3A—1 Zone; Phytotronics, Earth City, MO, USA). On 20 July, rooted cuttings were transplanted into 3.8 L pots containing a commercial growing substrate (Metro Mix® 360; SunGro, Bellevue, WA, USA). Until the start of the experiments, plants were irrigated with a water-soluble fertilizer solution (Peter's 15-5-15 Ca-Mg Special; Scotts, Marysville, OH, USA) at a nitrogen concentration of 150 ppm (16 mM), an electrical conductivity (EC) of 1.0 dS·m$^{-1}$, and a pH of 6.0. For insect control, all plants were top-dressed with 1% Marathon. During the experiments, in order to control powdery mildew, plants were alternately treated with triple action neem oil (Southern Agricultural Insecticides, Inc., Palmetto, FL, USA) at a rate of 1 mL per 130 mL and with soapy water, which was made by adding one teaspoon of olive oil and one tablespoon of liquid detergent to one gallon water. On 1 September 2017, uniform plants were selected, and treatment solutions were applied. On 23 October 2017, plants were harvested (Experiment 1). For Experiment 2, plants were transplanted on 15 September and harvested on 6 November 2017.

**Table 1.** A list of hydrangea cultivars irrigated with a nutrient solution or saline solution (EC = 5.0 dS·m$^{-1}$ (EC 5) or 10.0 dS·m$^{-1}$ (EC 10)) in two greenhouse experiments to determine relative salt tolerance.

| Species | Cultivar | Common Name |
| --- | --- | --- |
| *Hydrangea macrophylla* | 'Ayesha' | Bigleaf hydrangea |
| *H. macrophylla* | 'Emotion' | Bigleaf hydrangea |
| *H. macrophylla* | 'Mathilda Gutges' | Bigleaf hydrangea |
| *H. macrophylla* | 'Merritt's Supreme' | Bigleaf hydrangea |
| *H. macrophylla* | 'Passion' | YouMe® Bigleaf hydrangea |
| *H. paniculata* | 'Interhydia' | Pink Diamond hardy hydrangea |
| *H. paniculata* | 'Bulk' | Quickfire® panicle hydrangea |
| *H. quercifolia* | 'Snowflake' | Oakleaf hydrangea |
| *H. serrata* | 'Preciosa' | Mountain hydrangea |
| *H. serrata* × *macrophylla* | 'Sabrina' | Hybrid hydrangea |
| *H. serrata* × *macrophylla* | 'Selina' | Hybrid hydrangea |

### 2.2. Treatments

For both experiments, a total of three treatments (two saline solutions and a control solution) were used. All treatment solutions were prepared in 100 L tanks with reverse osmosis (RO) water and Peter's 15-5-15 Ca-Mg Special (Scotts, Marysville, OH, USA) at a nitrogen concentration of 150 ppm (16 mM), and pH adjusted to 6.0. The EC of the control solution was 1.0 dS·m$^{-1}$. Sodium chloride and calcium chloride were used to prepare the salt treatment solutions at an EC of 5.0 and 10.0 dS·m$^{-1}$ (EC 5 and EC 10 treatments). These salt levels were selected to represent moderate and severe saline irrigations to best assess salt-induced stress responses in plants. For all plants, 1.0 L of the respective treatment solutions were applied once a week, which resulted in an approximately 20% leaching fraction. Irrigation frequency depended on environmental conditions and plant biomass. For plants that needed irrigation more than once per week (usually the control plants and EC 5 treatment), nutrient solution was applied to avoid excessive salt accumulation in the root zone and stabilize leachate EC.

### 2.3. Data Collection

Leachate EC was collected weekly according to the methods described by Cavins et al. [17]. Briefly, 50 mL of RO water was applied to a randomly selected plant per cultivar per treatment approximately 15 min following irrigation, and the subsequent leachate was collected and EC and pH were measured using LaQua twin probes (Horiba, Japan). A visual score was adapted from Liu et al. [14] and was based on a visual assessment of salt damage (foliar necrosis) on a scale of 0 to 5 (0 = dead, 1 = 80% damage, 2 = 60% damage, 3 = 40% damage, 4 = 20% damage, and 5 = 0% damage, approximately).

At termination, the leaf area and shoot dry weight (DW) data were collected. Leaf area was measured using a LI-3100C area meter (LI-COR®, Biosciences, Lincoln, NE, USA). Shoot dry weight was measured after plant tissue was thoroughly dried in an oven at 60 °C for at least 3 days.

## 2.4. Leaf Mineral Analysis

Dried leaf samples were ground using a Wiley Mill (Thomas Scientific, Swedesboro, NJ, USA) to pass a 40-mesh screen. Dry powder samples were extracted with 2% acetic acid and chloride concentration was measured from the extracted solution using a chloride analyzer (M926; Cole Parmer Instrument Company, Vernon Hills, IL, USA), according to the methods described by Gavlak et al. [18]. Powder samples were also sent to the Soil, Water and Forage Testing Laboratory at Texas A&M University (College Station, TX, USA) for the mineral analysis of $Na^+$, $K^+$ and $Ca^{2+}$, according to methods described by Havlin and Soltanpour [19] and Issac and Johnson [20].

## 2.5. Greenhouse Location and Environment

The experiments were conducted at the Texas A&M AgriLife Research Center at El Paso (31°41′50.5″ N 106°16′56.8″ W) in a fan and pad evaporative-cooled greenhouse. The actual average greenhouse air temperatures were between 21.6 and 24.0 °C and the average daily relative humidity ranged from 44% to 55%. The average daily light integral was between 7.7 and 16.4 mol·m$^{-2}$·d$^{-1}$ during the experimental period.

## 2.6. Experimental Design and Statistical Analysis

A modified split-plot design was used with six replications ($N = 198$). Treatments were the main plot and cultivars were the sub plot. Cultivars were randomized within each main plot. A two-way analysis of variance (ANOVA) was used to test the effects of salinity and cultivar on plant growth. Means separation among treatments was conducted using Tukey's honest significant difference (HSD) multiple comparison test. All statistical analyses were carried out using JMP (Version 13.2; SAS Institute, Cary, NC, USA). Tissue mineral analysis was only carried out for Experiment 1.

## 3. Results and Discussion

### 3.1. Leachate EC

The average leachate EC during Experiment 1 ranged from 2.4 to 3.5 dS·m$^{-1}$ for the control (Figure 1A). For EC 5 and EC 10 treatments, leachate EC increased from 5.0 to 9.0 dS·m$^{-1}$ and from 6.7 to 14.2 dS·m$^{-1}$, respectively, in the first four weeks, and then stabilized during the last four weeks of the study. The leachate EC levels being higher than those of the irrigation water indicated salt accumulation in the root zone. Salt accumulation depends on the salinity level of irrigation water, leaching fraction, substrate type and irrigation frequency. In this study, we applied treatment solution once a week with an approximately 20% leaching fraction. The trends of leachate EC in all treatments agreed with those in our previous studies [14,21].

In Experiment 2, leachate EC ranged from 2.7 to 3.5 dS·m$^{-1}$, 5.5 to 9.8 dS·m$^{-1}$ and 7.0 to 17.3 dS·m$^{-1}$ in the control, EC 5 and EC 10 treatments, respectively. The leachate EC values in both experiments had similar trends over time. Salts are known to accumulate in the substrate without excessive leaching, and this difference may be due to variation in both the leaching fraction and evapotranspiration rates.

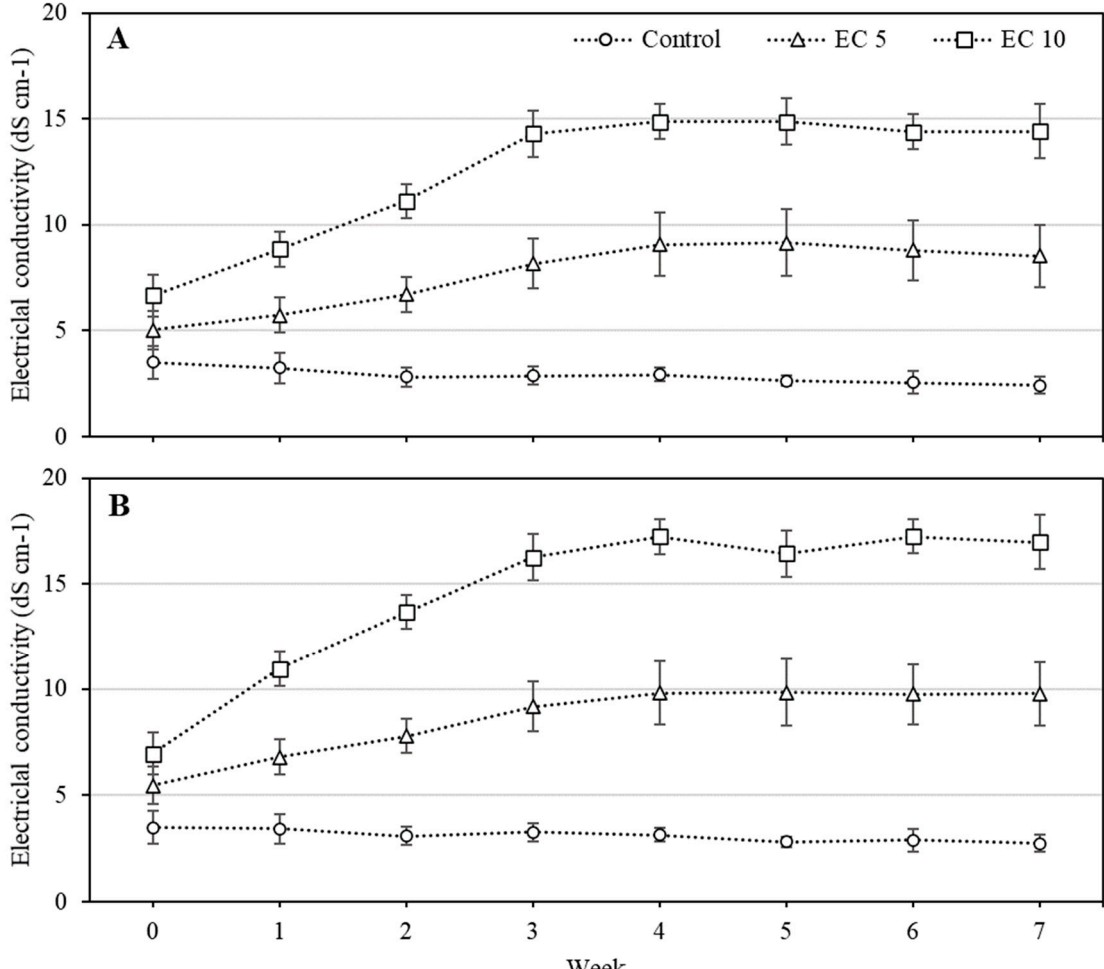

**Figure 1.** Leachate electrical conductivity (EC) in Experiment 1 (**A**) and Experiment 2 (**B**). Control represents a nutrient solution at an EC of 1.0 dS·m$^{-1}$; EC 5 represents a saline solution at an EC of 5.0 dS·m$^{-1}$; and EC 10 represents a saline solution at an EC of 10.0 dS·m$^{-1}$. Vertical bars represent standard deviation.

### 3.2. Visual Quality

There were significant interactions between treatment and cultivar for visual quality, leaf area and shoot DW, indicating that the cultivars responded to treatment differently (Table 2). In the EC 10 treatment, most plants of *H. paniculata* 'Interhydia' and 'Bulk' were dead, with the lowest visual scores of 0.6 and 0.5 (Table 2), respectively. By the fourth week of the experiment, some plants of these two cultivars in the EC 10 treatment were already dead, or exhibited severe salt damages with few green leaves, indicating that these two cultivars were the least salt-tolerant. Other cultivars with lower visual scores were 'Preciosa', 'Passion' and 'Snowflake'. 'Ayesha' and 'Merritt's Supreme', and the two hybrids 'Sabrina' and 'Selina', had relatively higher visual scores. In the EC 5 treatment, 'Bulk' had the lowest score of 1.1, followed by 'Preciosa' and 'Passion' with 2.4 and 2.8. All other cultivars had visual scores of 3.4 or higher.

In the second experiment, again, most plants of 'Bulk' and 'Interhydia' in EC 10 and 'Bulk' in EC 5 were dead or severely damaged by the end of the experiment (Table 2). The next group with poor visual quality included 'Passion', 'Preciosa', 'Emotion', and 'Snowflake' with scores below 2.0. Although there were some differences in visual scores between Experiments 1 and 2, the trends and relative order in salt tolerance were generally similar.

**Table 2.** Mean visual score, leaf area and total shoot dry weight (DW) of 11 hydrangea cultivars irrigated with a nutrient solution (electrical conductivity (EC) = 1.0 dS·m$^{-1}$; control) or a saline solution (EC = 5.0 dS·m$^{-1}$ (EC 5) or 10.0 dS·m$^{-1}$ (EC 10)) in Experiments 1 and 2. Significance of main effects and interactions denoted by ***, ** and * ($p < 0.001$, $p < 0.01$, and $p < 0.05$, respectively).

| Cultivar | Visual Score [z] | | | Leaf Area (cm$^2$) | | | Shoot DW (g) | | |
|---|---|---|---|---|---|---|---|---|---|
| | Control | EC 5 | EC 10 | Control | EC 5 | EC 10 | Control | EC 5 | EC 10 |
| Experiment 1 | | | | | | | | | |
| Ayesha | 4.7 Aa [y] | 3.9 Ab | 3.8 ABb | 3129 a | 1635 b | 1315 b | 26.5 Aa | 15.9 Ab | 13.9 Ab |
| Emotion | 4.0 Ba | 3.6 Ab | 2.1 Dc | 1675 a | 1166 b | 466 c | 15.6 BCa | 11.6 ABCa | 5.4 Bb |
| Mathilda Gutges | 4.5 ABa | 3.9 Ab | 2.8 Cc | 1674 a | 1158 b | 575 c | 16.7 BCa | 11.4 ABCb | 6.3 Bc |
| Merritt's Supreme | 4.3 ABa | 3.7 Ab | 3.2 BCb | 1602 a | 978 b | 652 c | 13.7 BCDEa | 9.2 BCDb | 6.9 Bb |
| Passion | 4.1 ABa | 2.8 BCb | 0.8 Ec | 1838 a | 1080 b | 206 c | 16.8 BCa | 13.7 ABa | 4.7 Bb |
| Interhydia | 4.5 ABa | 3.6 Ab | 0.6 Ec | 1228 a | 795 b | - [x] | 15.2 BCDa | 9.2 BCDb | 6.0 Bb |
| Bulk | 3.0 Ca | 1.1 Db | 0.5 Ec | 507 a | 234 b | - | 6.7 Ea | 4.2 Da | 4.0 Ba |
| Snowflake | 4.4 ABa | 4.1 Aa | 0.6 Eb | 2211 a | 934 b | 814 b | 17.9 Ba | 7.8 CDb | 5.5 Bb |
| Preciosa | 3.1 Ca | 2.4 Cb | 0.7 Ec | 984 a | 514 b | 150 c | 8.0 DEa | 4.8 Db | 3.6 Bb |
| Sabrina | 3.9 Ba | 3.8 Aa | 4.0 Aa | 2257 a | 1528 b | 1205 b | 20.1 ABa | 14.6 CDb | 11.5 Ab |
| Selina | 3.2 Ca | 3.4 ABa | 3.5 ABa | 1250 a | 1003 ab | 726 b | 8.7 CDEa | 8.5 CDa | 7.0 Ba |
| Treatment (T) | *** | | | *** | | | *** | | |
| Cultivar (C) | *** | | | *** | | | *** | | |
| T × C | *** | | | ** | | | *** | | |
| Experiment 2 | | | | | | | | | |
| Ayesha | 4.0 ABCa [y] | 4.0 Aa | 3.3 Ab | 2604 a | 1996 b | 1333 c | 23.1 ABa | 21.7 Aa | 14.9 Ab |
| Emotion | 3.5 BCDa | 2.5 DEb | 1.7 CDc | 1353 a | 947 a | 187 b | 14.8 CDa | 11.7 BCa | 10.6 ABCa |
| Mathilda Gutges | 4.0 ABCa | 3.5 ABCa | 2.3 BCb | 1305 a | 1064 a | 531 b | 14.8 CDa | 10.2 BCDb | 7.2 CDc |
| Merritt's Supreme | 3.4 BCDa | 3.2 ABCDa | 2.6 ABCa | 1429 a | 843 b | 724 b | 12.8 CDEa | 8.9 BCDEa | 10.3 ABCa |
| Passion | 3.3 CDa | 2.0 Eb | 1.0 DEFc | 1462 a | 934 b | - [x] | 15.4 BCDa | 13.1 BCb | 4.4 Dc |
| Interhydia | 4.2 ABa | 2.7 CDEb | 0.5 Fc | 1182 a | 573 b | - | 13.8 CDa | 7.9 CDEab | 6.9 CDb |
| Bulk | 3.0 Da | 0.8 Fb | 0.5 EFb | 315 | - | - | 5.8 Ea | 4.4 DEa | 4.1 CDa |
| Snowflake | 4.7 Aa | 3.7 ABa | 1.8 CDb | 2513 a | 1598 ab | 905 b | 24.0 Aa | 14.7 Bab | 9.1 BCDb |
| Preciosa | 3.0 Da | 2.2 Eb | 1.4 DEc | 831 a | 352 b | 209 b | 8.4 DEa | 4.0 Eb | 4.3 Db |
| Sabrina | 3.0 Db | 3.7 ABa | 3.0 ABb | 1966 a | 1496 b | 1136 c | 16.9 ABCa | 14.0 Bab | 12.3 ABb |
| Selina | 3.5 BCDa | 2.8 BCDEa | 2.8 ABa | 1166 a | 824 b | 549 b | 10.6 CDEa | 7.9 CDEb | 6.4 CDb |
| Treatment (T) | *** | | | *** | | | *** | | |
| Cultivar (C) | *** | | | *** | | | *** | | |
| T × C | *** | | | *** | | | *** | | |

[z] 0 = dead, 1 = 80% damage, 2 = 60% damage, 3 = 40% damage, 4 = 20% damage and 5 = 0% damage, approximately. [y] For each metric, means followed by different letters indicate significant difference according to Tukey's Honestly Significant Difference (HSD) test at $\alpha$ = 0.05; uppercase among cultivars; lowercase among treatments. [x] All plants died or were severely damaged by salt stress.

Conolly et al. [13] evaluated the relative salt tolerance of five hydrangea species based on percent necrotic leaf area. They treated the plants using a foliar spray with full-strength seawater (ion concentration approximate to seawater) salt solution, half-strength seawater solution, or tap water. They found that cultivars of *H. macrophylla* and *H. serrata* were more tolerant of full-strength spray than those of *H. paniculata*, *H. anomala* and *H. arborescens*. They also observed different responses to the half-strength spray, wherein *H. anomala* ssp. *petiolaris* was most tolerant and *H. macrophylla* and *H. serrata* were the second most tolerant, indicating interactive effects between salt spray concentration and species. While our study was different from that of Conolly et al. [13] in treatment methods, our results confirmed that *H. paniculata* cultivars were not salt-tolerant. Further, we observed that some cultivars in *H. macrophylla* were relatively salt-tolerant, but 'Passion' was not, indicating variation

among cultivars of the same species. As for *H. serrata*, we only tested one cultivar, 'Preciosa', which was the second least salt-tolerant based on visual quality.

Foliar discoloration and leaf edge browning were observed in almost all cultivars for unknown reasons, which is why plants in the control treatment did not get a visual score of 5. Some cultivars suffered powdery mildew, which further lowered the visual score. For ornamental plants, aesthetic appearance is of great importance [22]. Healthy foliage and abundant flowers with adequate size are important ornamental traits. Because this study was conducted during the vegetative growth stage, the impact of salt stress on flower performance could not be directly evaluated. However, the number of flowers per plant is directly correlated with healthy vegetative growth, such as in roses [23]. In salt-sensitive cultivars, vegetative growth is significantly reduced, and the magnitude of reduction is dependent on stress level and exposure time. Miralles et al. [2] reported that irrigating *H. macrophylla* 'Leuchtfeurer' with saline water at an EC of 5.65 dS·m$^{-1}$ (leachate EC of around 10 dS·m$^{-1}$) reduced all the aerial organs, including leaf, inflorescence and flower.

Based on the visual scores and early mortality in the middle of the study, the two cultivars of 'Bulk' and 'Interhydia' were the least tolerant to salt stress. The next least tolerant cultivar was 'Preciosa', followed by 'Passion'. The most salt-tolerant among all 11 cultivars was 'Ayesha', followed by the two hybrids 'Sabrina' and 'Selina'. 'Mathilda Gutges', 'Merritt's Supreme' and 'Snowflake' were moderately tolerant.

### 3.3. Plant Growth

In Experiment 1, elevated salinities (both EC 5 and EC 10) reduced leaf area in all cultivars, with reductions ranging from 20% to 58% in EC 5 and 42% to 89% in EC 10 treatments (Table 2). Total shoot DW was also reduced, although there were no statistical differences between the control and EC 5 or EC 10 in some cultivars, due to the high variation and mortality in these cultivars. In Experiment 2, the leaf area was reduced in most cultivars, with no statistical difference between the control and EC 5 treatment in the cultivars 'Emotion', 'Mathilda Gutges' and 'Snowflake' (Table 2). Total shoot DW was reduced in all cultivars except for 'Emotion', 'Mathilda Gutges' and 'Bulk'.

Growth reduction percentage is another important parameter used to evaluate relative salt tolerance among different cultivars of ornamental plants [2,3,14,16,21,24,25]. Miralles et al. [2] reported a reduction in shoot DW of 73%, and leaf area of 68%, in *H. macrophylla* 'Leuchtfeurer', when irrigated with saline water at an EC of 5.65 dS·m$^{-1}$ (leachate EC of around 10 dS·m$^{-1}$) for 11 weeks, compared to control (irrigated with fresh water). These results indicated that the growth of *H. macrophylla* plants was significantly reduced when irrigated with water at elevated salinity. Significant vegetative growth reduction might lead to lower flower numbers. Therefore, future studies should be conducted in the spring growing season to quantify the impact of salinity on the flowering characteristics of these hydrangea cultivars.

### 3.4. Mineral Analysis

There were significant effects of treatment and cultivar, and interactive effects between treatment and cultivar, on leaf concentrations of Na$^+$, Cl$^-$, Ca$^{2+}$ and K$^+$ (Table 3). Elevated salinities increased the concentrations of leaf Na$^+$ and Cl$^-$ in all cultivars compared to those of the control. The EC 5 treatment increased the leaf Na$^+$ concentration by 3 times in 'Merritt's Supreme' and by 22 times in 'Snowflake'. The highest Na$^+$ concentration in the EC 5 treatment was found in 'Passion' (19.0 mg g$^{-1}$), followed by 'Emotion' (12.8 mg g$^{-1}$), and then by 'Sabrina' and 'Selina' (9.7 mg g$^{-1}$). In the EC 10 treatment, the highest leaf Na$^+$ concentrations were found in 'Passion' (35.8 mg g$^{-1}$) and 'Emotion' (31.6 mg g$^{-1}$), which correlated with those in the EC 5 treatment, the highest among all cultivars. The next highest Na$^+$ concentrations were found in 'Preciosa', 'Sabrina' and 'Selina'. For the cultivars 'Bulk' and 'Interhydia', leaf Na$^+$ concentrations might have been underestimated as the plants were severely affected, and some even died before the end of the experiment. The most salt-tolerant cultivar,

'Ayesha', had relatively low $Na^+$ concentrations compared to other cultivars. Notably, the cultivars 'Snowflake' and 'Merritt's Supreme' also had low leaf $Na^+$ concentrations.

For leaf $Cl^-$ concentration, similar to leaf $Na^+$ concentration, the highest increases compared to their respective controls were found with 'Interhydia', 'Bulk', 'Snowflake' and 'Passion'. In the EC 5 and EC 10 treatments, the leaf $Cl^-$ concentration increased by 11 and 20 times, respectively, in 'Passion', and 59 and 101 times, respectively, in 'Interhydia'. Among the control plants, the highest $Na^+$ concentrations were observed in 'Preciosa', 'Sabrina', 'Merritt's Supreme', 'Mathilda Gutges', 'Passion', 'Emotion' and 'Selina', while the lowest $Na^+$ concentrations were found in 'Interhydia', 'Bulk' and 'Snowflake'. As regards leaf $Cl^-$ concentration, the highest was found in 'Preciosa' at 10.3 mg $g^{-1}$, followed by 'Sabrina' at 8.1 mg $g^{-1}$. Similar to leaf $Na^+$ concentration, the lowest $Cl^-$ concentrations were found in 'Interhydia', 'Bulk' and 'Snowflake', the three cultivars with the lowest $Na^+$ concentrations. These results indicated that the salt-sensitive cultivars have inherently low tolerance to harmful ions of $Na^+$ and $Cl^-$, which is why they did not survive under high tissue $Na^+$ and $Cl^-$ accumulation. For example, *Monarda citriodora*, a wildflower, showed low tissue tolerance to $Na^+$ concentration, with 100% mortality when irrigated with saline solution at an EC as low as 2.8 dS·m$^{-1}$ [25].

For leaf $Ca^{2+}$ concentration, except for the cultivar 'Bulk', elevated salinity treatments increased leaf $Ca^{2+}$ concentration. In the EC 5 treatment, the increase in $Ca^{2+}$ concentration compared to the control ranged from 15% in 'Sabrina' to 80% in 'Snowflake', and 58% in 'Sabrina' to 120% in 'Snowflake' in the EC 10 treatment. It is known that $Ca^{2+}$ influx into the cytoplasm is a plant response to environmental stresses, including salt stress [26]. Calcium has been shown to slow down the accumulation of $Na^+$ within the plant and mitigate the detrimental effects of elevated $Na^+$, particularly in the leaf tissue [27,28].

For leaf $K^+$ concentration, there were minor but significant decreases in 'Selina' and 'Passion', and increases in 'Emotion', 'Merritt's Supreme' and 'Interhydia' when treated with EC 10, compared to the control (Table 3). The maintenance of high $K^+$ in the shoot tissue has been noted as a salt-tolerant trait in some plant species due to its ability to aid in osmotic adjustment [29]. However, in this study, no clear trends were identified with respect to leaf $K^+$ concentration and salt tolerance.

Although $Na^+$ is not an essential element for plants, it is considered a beneficial element to some halophyte species [30] and can also be used as a micronutrient to aid in carbon fixation in C4 and CAM (crassulacean acid metabolism) plants [31]. In some plants, $Na^+$ can also be used as a partial replacement of $K^+$ as an osmoticum [32]. In contrast, $Cl^-$ is an essential plant nutrient, and is needed in small quantities as a micronutrient aiding in plant metabolism, photosynthesis, osmosis and ionic balance within cells [33]. The typical leaf $Na^+$ concentration for glycophytes is between 0.2 and 2.0 mg $g^{-1}$ [34], which is the range observed in the control treatment in the present study, with an average of 1.1 mg $g^{-1}$ across all cultivars. Leaf $Cl^-$ concentration is generally higher than that of $Na^+$ due to its mobility in plants [33], and in this study the average was 4.5 mg $g^{-1}$ across all cultivars, ranging from 0.6 to 10.3 mg $g^{-1}$ (Table 3).

**Table 3.** Mean leaf sodium ($Na^+$), chloride ($Cl^-$), calcium ($Ca^{2+}$) and potassium ($K^+$) concentrations of 11 hydrangea cultivars irrigated with a nutrient solution (electrical conductivity (EC) = 1.0 dS·m$^{-1}$; control) or a saline solution (EC = 5.0 dS·m$^{-1}$ (EC 5) or 10.0 dS·m$^{-1}$ (EC 10)) in Experiment 1. Significance of main effects and interactions denoted by ***, ** and * ($p < 0.001$, $p < 0.01$ and $p < 0.05$, respectively).

| Cultivar | $Na^+$ (mg·g$^{-1}$) | | | $Cl^-$ (mg·g$^{-1}$) | | | $Ca^{2+}$ (mg·g$^{-1}$) | | | $K^+$ (mg·g$^{-1}$) | | |
|---|---|---|---|---|---|---|---|---|---|---|---|---|
| | Control | EC 5 | EC 10 | Control | EC 5 | EC 10 | Control | EC 5 | EC 10 | Control | EC 5 | EC 10 |
| Ayesha | 0.8 Cc [z] | 6.2 CDb | 9.7 CDa | 3.3 Dc | 30.3 CDb | 44.2 CDEa | 11.6 Eb | 19.8 CDa | 22.0 Ca | 39.5 BCa | 39.4 Ba | 40.0 ABCDa |
| Emotion | 1.5 Ac | 12.8 Bb | 31.6 Aa | 4.4 Dc | 46.2 ABb | 92.1 Aa | 18.7 CDc | 28.7 CDb | 37.8 ABa | 33.4 DEFb | 34.1 BCDab | 37.3 BCDa |
| Mathilda Gutges | 0.9 Cb | 5.6 CDb | 14.8 BCa | 5.2 CDc | 32.9 BCDb | 65.2 Ba | 16.8 CDEc | 23.6 CDb | 31.4 BCa | 39.7 BCa | 40.3 Ba | 43.8 ABa |
| Merritt's Supreme | 1.5 ABc | 6.0 CDb | 10.5 BCDa | 6.9 BCc | 31.1 CDb | 49.2 BCDEa | 16.8 CDEb | 21.2 CDb | 29.9 BCa | 37.2 BCDb | 38.2 BCb | 42.1 ABCa |
| Passion | 1.7 Ac | 19.0 Ab | 35.8 Aa | 4.6 Dc | 54.8 Ab | 96.7 Aa | 21.2 BCb | 31.6 BCa | 33.0 BCa | 38.3 BCDa | 35.9 BCDab | 33.2 Db |
| Interhydia | 0.2 Db | 3.3 Db | 16.9 BCa | 0.6 Ec | 36.2 BCb | 61.4 BCa | 26.6 Bc | 41.9 ABb | 47.6 Aa | 29.0 EFb | 29.2 Dab | 35.1 CDa |
| Bulk | 0.4 CDa | 2.0 Da | 5.6 Da | 1.1 Eb | 28.2 CDa | 31.2 Ea | 34.0 Aa | 45.0 Aa | 38.1 ABa | 28.6 Fa | 34.5 BCDa | 33.8 CDa |
| Snowflake | 0.1 Db | 2.3 Db | 11.6 BCDa | 0.9 Ec | 20.8 Db | 55.2 BCDa | 15.5 CDEc | 28.1 CDb | 34.2 BCa | 34.5 CDEab | 31.5 CDb | 39.2 BCDa |
| Preciosa | 1.0 BCb | 2.9 Db | 16.8 BCa | 10.3 Ac | 28.9 CDb | 66.4 Ba | 18.1 CDc | 22.8 CDb | 29.8 BCa | 51.0 Aa | 51.2 Aa | 48.4 Aa |
| Sabrina | 1.8 Ac | 9.7 BCb | 18.8 Ba | 8.1 Bc | 32.6 BCDb | 60.5 BCDa | 14.6 DEb | 16.8 Db | 22.8 Ca | 40.9 Ba | 36.9 BCa | 37.3 BCDa |
| Selina | 1.7 Ac | 9.7 BCb | 17.4 BCa | 4.0 Dc | 21.7 CDb | 42.3 DEa | 13.2 DEb | 16.6 Dab | 21.7 Ca | 49.2 Aa | 40.4 Bb | 39.1 BCDb |
| Treatment (T) | *** | | | *** | | | *** | | | * | | |
| Cultivar (C) | *** | | | *** | | | *** | | | *** | | |
| T × C | *** | | | *** | | | * | | | *** | | |

[z] For each metric, means followed by different letters indicate significant differences according to Tukey's Honestly Significant Difference (HSD) test at $\alpha = 0.05$; uppercase among cultivars; lowercase among treatments.

Two of the important mechanisms of salt tolerance are the exclusion of $Na^+$ and $Cl^-$ in the leaf tissue, and the tolerance of high tissue concentrations of $Na^+$ and $Cl^-$ [35]. $Na^+$ and $Cl^-$ at high concentrations are harmful to healthy plant growth in glycophytes. In many cases, salt-tolerant genotypes have lower leaf $Na^+$ and/or $Cl^-$ concentrations in leaves and stems, and this is achieved by preventing these ions from entering shoots. In other cases, plants tolerate high $Na^+$ and $Cl^-$ concentrations by partitioning $Na^+$ and $Cl^-$ in the cell vacuole [1,24,35]. For example, among 10 aster species, *Eupatorium greggii* and *Viguiera stenoloba* were relatively salt-tolerant with lower leaf $Na^+$ concentrations, while another relatively salt-tolerant species, *Santolina chamaecyparissus*, had higher leaf $Na^+$ concentration [16]. In cereal crops, sorghum, which is salt-tolerant, had very low $Na^+$ and $Cl^-$ concentrations in leaves and stems, while maize genotypes had much higher $Na^+$ and $Cl^-$ concentrations in leaves and stems at elevated salinity levels [36].

Large variations were observed in the current study regarding the relative salt tolerance among the four species and one hybrid of hydrangea (*H. macrophylla*, *H. paniculata*, *H. quercifolia*, *H. serrata* and *H. serrata* × *macrophylla*), and within cultivars of *H. macrophylla* based on growth reduction and leaf $Na^+$ and $Cl^-$ concentrations. For example, *H. macrophylla* 'Passion' showed poor performance, with severe foliar salt damage, high mortality and greater reductions in leaf area and shoot DW compared to other cultivars in the same species. In our previous study with *H. macrophylla* 'Smhmtau' and 'Smnhmsigma', plants were treated with the same saline solutions at an EC of 5.0 dS·m$^{-1}$ or 10.0 dS·m$^{-1}$ for 35 days; however, the leaf $Na^+$ and $Cl^-$ concentrations were higher than those observed in the current study [14]. The salt tolerance of plants largely varies among species and even among cultivars of the same species, as evidenced in the current study.

Leaf $Na^+$ and $Cl^-$ concentrations have been shown to be negatively correlated with plant growth [36–38]. In this study, there were similar trends between shoot DW and leaf $Na^+$ and $Cl^-$ concentrations ($Na^+$ vs. DW: $p$ =0.034; $Cl^-$ vs. DW: $p$ = 0.003). Similarly, visual scores (VS) were negatively correlated with leaf $Na^+$ and $Cl^-$ concentrations, and with relative shoot growth (VS vs. $Na^+$, $p$ = 0.0111; VS. vs. $Cl^-$, $p$ = 0.0012; VS vs. DW: $p$ < 0.0001). That is, the greater the accumulation of $Na^+$ and $Cl^-$ in leaves, the lower the visual quality ratings and lesser the growth that occurred.

## 4. Conclusions

Among the 11 cultivars of four species and one hybrid of hydrangea, based on visual quality and growth reduction, we concluded that 'Ayesha' (*H. macrophylla*), and the two hybrids 'Sabrina' and 'Selina', were the most tolerant, and a second tolerant group included 'Mathilda Gutges' and 'Merritt's Supreme' (*H. macrophylla*). The next most salt-tolerant cultivars included 'Preciosa' (*H. serrata*) and 'Passion' (*H. macrophylla*), followed by 'Snowflake' (*H. quercifolia*) and 'Emotion' (*H. macrophyalla*). 'Bulk' of *H. paniculata* was the least salt-tolerant, followed by 'Interhydia' (*H. paniculata*). The salt-sensitive cultivars 'Bulk', 'Interhydia' and 'Snowflake' had inherently low leaf $Na^+$ and $Cl^-$ concentrations. Salt tolerance varied with species and cultivars within the species *H. macrophylla*.

**Author Contributions:** G.N., Y.S., and T.H. conceptualized and designed the study. T.H., H.D. and C.P. performed the experiments and collected data; Y.S. analyzed data. G.N. drafted the manuscript; T.H., Y.S. and J.A. revised. All authors have read and agreed to the published version of the manuscript.

**Funding:** This research was supported in part by the U.S. Department of Agriculture (USDA) National Institute of Food and Agriculture Hatch Project TEX090450 and Floricultural and Nursery Research Initiative.

**Acknowledgments:** The authors would like to thank Oregon Hydrangea Company (Brookings, OR, USA) for donating hydrangea cuttings.

**Conflicts of Interest:** The authors declare no conflict of interest.

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
