# Peer review of "Salt Tolerance of Hydrangea Plants Varied among Species and Cultivar within a Species"

_horticulturae, doi:10.3390/horticulturae6030054_

Round 1

Reviewer 1 Report

Dear Authors, I'm sorry but I'm very doubtful about this work. It appears as a basic work on salt stress with very poor considerations/data on physiological responses that normally are used in this kind of work (e.g., gas exchange analysis, fv/fm ratio, chlorophyll content, ecc.). 

Why did you decide to carry out these two experiments in fall summer, early autumn? The tested species are widespread on both cut flower and bush ornamental market and flowering characterisitcs as well as effects on flower (number and quality) are very important parameters to be evalueted (especially when cvs resulted as salt tolerant or moderately sensitive).

Moreover you showed some pictures in fig. 2 where for many cvs. many problems are clearly detectables in control treatments, so it's very difficult to understand when cvs. resulted really salt tolerant. 

In mineral analysis you did not almost comments obtained results about K+ and Ca++ that you measured and showed in table 2. 

Other comments are showed directly in the attached document. 

Kind regards.

Reviewer 2 Report

Review of Manuscript ID: horticulturae-887263 

Title: Salt tolerance of Hydrangea plants varied with species and cultivar within a species.

Summary of the manuscript:

The article describes the salt tolerance levels of cultivars of hydrangea, uses phenotypical changes to group them into sensitive and tolerant species/cultivars and the concentrations of sodium and chloride ions to predict the mechanism of tolerance if the different species. 

Review summary:

Major issues

The manuscript has a simple theme and is easy to read; the authors are commended for that.  However, scientific rigour has been sacrificed in the authors’ quest for simplicity.  The main issue with the manuscript is that the full results of the two experiments were not presented.  The authors’ decision to selectively report the results of one of the two experiments because the data were similar goes against the principles of conducting experiments with replicates (lines 138-141).  Scientifically, data from two experiments are meant to be reported whether they are similar or not.  Indeed, if the data from one experiment could represent the other (which is never scientifically accurate), then why not find means and report them with statistical and experimental errors so readers can judge how credible conclusions can be drawn from such data.  This decision is fallible and should be corrected.  To make the conclusions drawn from the experiments credible, the data from both experiments should be reported and where possible means should be calculated, and the number of replicates used for each reported data indicated.

The other notable change that needs to be done relates to the research question, as stated on lines 54 and 74.  The literature review following the sentence on line 54 indicates that some hydrangeas are salt-tolerant.  Why do the authors still think that “very limited information” is available?  How much more information is needed to make the point?  I, therefore, don't agree with the words "very limited".  The same goes for the sentence on line 74.  I think the authors will need to re-frame the research question in this manuscript.  The least they could say is that no information is available for the particular species and cultivars used in the study.  

Minor issues that still need attention:

  1. Title: The title of the manuscript could be written as “The level of salt tolerance varies between species and within cultivars of the same species of hydrangeas”.
  2. English language: The manuscript could benefit from English language editing. A few of these have been indicated in the pdf attached to the review.  Specific attention should be given to but not limited to 
    1. The order of words
    2. The use of present and past tenses; particularly all of the results and conclusions drawn from the results of the research MUST be in the past
  3. Presentation of Results: The data for some reported observations and parameters must be reported to give credibility to the research. Examples, on lines 231-233
  4. Analysis of data: Results of some statistical analyses must also be shown to support the statements made, for example, on lines 231-233 at what alpha level and on lines 307 and 308, correlation coefficients should be indicated and the methods for analysing those noted in the methods.
  5. Ambiguities in sentences: As part of making the whole manuscript easy to read (see point 2), the sentences on the following lines must be changed to make the results clear to the reader
    1. Lines 274-275
  6. Erroneous statements: Conclusions mustn't be predicted, without any scientific basis. An example is on lines 254 -257. 
  7. Any other comment on the attached pdf file of the manuscript should be considered.

Reviewer 3 Report

Comments to the Author

The authors screened 11 varieties of Hydrangea spp, using two levels of Salinity plus control in two independent greenhouse experiments. The outcome of the research is useful and significant to the Hydrangea growers, especially to establish farms in elevated saline soil or in using reclaimed water. The paper is written nicely with clear objectives.

However, the main limitation in the manuscript is in statistical analysis and must be improved. Results form both experiments should be included after combined analysis or using Exp as a fixed factor. I have highlighted issues in the attached PDF itself. Please find the attachment.

Thank you

Round 2

Reviewer 1 Report

Dear Authours, the revision version appears as more clear and complete. 

Nevertheless I strongly reccomend to remove the Figure 2 because it can bring to some misunderstanding in the reader as some control plants are very worse in quality than salt treatments (e.g., in snowflakes control seems to be traded place with EC5 treatment) and some visible symptoms are attributable also to salt or anyway phytotoxic stress (e.g., in 'Selina'). Moreover, in my opinion, in table 2, DW% must be shown, or also root to shoot ratio if available. 

I suggest also to specify in the text why you chose the autumn to conduct this first screening on salt stress suggesting the opportunity of further investigastion on salt tolerant or less sensitive cultivars about effects on flowering.

Finally, I think that in such experiment more deepened analysis on eco-physiological responses are very useful, in addition to give more scientific consistency to the manuscript. If some analysis are not possible on some treatments, a selection of salt tolerant or less sensitive cultivars could be anyway chose. If these kind of analysis are available (e.g., gas exchange, fv/fm, chl content...) I think you have to taken into account the opportunity to shown some data.

Kind regard

Reviewer 2 Report

Second Review of Manuscript ID: horticulturae-887263 

Title: Salt tolerance of Hydrangea plants varied with species and cultivar within a species.

Summary of the manuscript:

The article describes the salt tolerance levels of cultivars of hydrangea, uses phenotypical changes to group them into sensitive and tolerant species/cultivars and the concentrations of sodium, chloride, potassium and calcium ions to predict the mechanism of tolerance of the different species. 

Summary of Authors’ review:

The authors have attended to all the comments from the first review and this has improved the manuscript.  A few more questions have arisen from the improvements that need to be corrected to make the manuscript more scientific. 

Major issues

  1. Question on numbers: The number of replicated plants for each species used to obtain any of the data presented is not clearly stated in the manuscript. The statement on line 120 seem to suggest that one plant per cultivar was used to obtain data for each parameter assessed and yet means with standard deviations have been calculated.  Please make this clear.  
  2. Use of scientific and statistical terminology:
    1. Line 167: It is not scientifically correct to describe data from two experiments as “similar”. A more scientific approach is to test the differences between the means and state whether or not they are significantly different at a certain alpha level. 
    2. Lines 244-246: Statistically, 50% and 58% cannot be referred to as small percentages.
  3. Any other comment on the attached pdf file of the manuscript should be considered.

Reviewer 3 Report

The authors have edited and addressed some of the issues and comments. I am confident in the novelty of research findings and its positive impact. 

However, I still saw some flaws in statistical analysis. No significant tests were given to average leachate EC and also not discussed finding adequately.

It was mentioned that Leaf sodium (Na+) and chloride (Cl–) concentrations were negatively correlated with visual quality, leaf area, and shoot DW---- where is your analysis, is that significant? etc.

Where is analysis for the time factor?

The interaction effects were highly/significant in both experiments but have described anywhere in result or discussion.

I highly recommend consulting with a statistician and revision of the manuscript before full acceptance.

Other comments:

Give standard error in Fig 1.

Rearrange Fig. 2 showing major groups of:

  1. Most salt-tolerant to
  2. Least salt-tolerant/salt-sensitive cultivars

Also, define particular tolerant groups and be consistent in writing. Rewrite the conclusion section by grouping them into a highly tolerant, tolerant, and sensitive group, etc. For example-- for cultivar "Bulk" mention it as salt-sensitive or as least tolerant across the manusript...dont interchange....it is confusing etc.

By convention, try to give positive results first, ie most tolerant followed by least-tolerant or sensitive cultivars, etc.

Thanks 

Round 3

Reviewer 3 Report

Dear Authors, 

Thank you for your edits and corrections. It is up to you whether to combine the experiment and consider the time factor or not. This manuscript still needs statistical backup, such as correlation analysis (as you mention it in the abstract as well), experimental design-wise analysis to track the main and sub-plot effect. However, I guess the finding will remain the same in the way you have analyzed it. 

I suggest going through English proof mainly in the use of present or past tense. Some sentences are hard to understand and synthesized incorrectly. There are multiple spaces between words, as well.

I highlighted a few in the attached pdf.

Thank you

Author Response

Thank you for your tireless review and critics of our manuscript.
Thank you for your edits and corrections. It is up to you whether to combine the experiment and consider the time factor or not. This manuscript still needs statistical backup, such as correlation analysis (as you mention it in the abstract as well), experimental design-wise analysis to track the main and sub-plot effect. However, I guess the finding will remain the same in the way you have analyzed it.
We have added the statistical analysis results (the probability values in the sentences). Please see the following:
In this study, we also found similar trends between shoot DW and leaf Na+ and leaf Cl– concentrations (Na+ vs. DW: P =0.034; Cl– vs. DW: P = 0.003). Similarly, visual scores (VS) were negatively correlated with leaf Na+ and Cl– concentrations and with relative shoot growth (VS vs. DW: P< 0.0001; VS vs. Na+, P = 0.0111; VS. vs. Cl–, P = 0.0012).
The reason we did not add these probability values is because it is the general trend we have found in our research. We did backup our statement with several references including a review paper of ours. In the Discussion section we consider we can write statements like that without giving statistical results. We do prefer to add statistical proof when results are presented.
I suggest going through English proof mainly in the use of present or past tense. Some sentences are hard to understand and synthesized incorrectly. There are multiple spaces between words, as well.
As for English proof, I am afraid, I don’t agree with the reviewer in many places. For examples:
Line 222: we used present tense because this is the footnote for the table. It is not the description of the results.
Line 240: we used present tense because this is general knowledge or other researcher’s work. It is not description of our results.
Line 233: Reviewer wanted the following: “…Total shoot DW reduced in all cultivars except for …”
Our original sentence: “…Total shoot DW was reduced in all cultivars except for …”
That is, reviewer wanted to delete “was”. I don’t agree. Throughout the “Results” section, we use passive sentences to describe these results, such as “…. leaf area was reduced …” instead of “leaf area reduced …” To be consistent, we did not want to make any change in these sentences suggested by the reviewer.
Line 236:
Our sentence is “Miralles et al. [2] reported a reduction in shoot DW of 73% and leaf area of 68%, compared to control ….”
Reviewer wanted to replace “of” with “by” …. Sorry, we can’t follow reviewer’s suggestion.
Also, reviewer indicated that there are multiple spaces between words. This is simply not true. I assume the reviewer meant extra space in between sentences. We did intentionally leave two spaces in between sentences. This is because when auto justification is used, some sentences are too close to each other. Nevertheless, in this revised version, we have deleted the extra space.
